# Salivary Biomarkers of Stress, Anxiety and Depression

**DOI:** 10.3390/jcm10030517

**Published:** 2021-02-01

**Authors:** Sylwia Chojnowska, Iwona Ptaszyńska-Sarosiek, Alina Kępka, Małgorzata Knaś, Napoleon Waszkiewicz

**Affiliations:** 1Faculty of Health Sciences, Lomza State University of Applied Sciences, Akademicka Street 14, 18-400 Lomza, Poland; mknas@pwsip.edu.pl; 2Department of Forensic Medicine, Medical University of Bialystok, 15-269 Białystok, Poland; i.sarosiek@wp.pl; 3Department of Biochemistry, Radioimmunology and Experimental Medicine, The Children’s Memorial Health Institute, 04-730 Warsaw, Poland; a.kepka@ipczd.pl; 4Department of Psychiatry, Medical University of Bialystok, 16-070 Choroszcz, Poland; napwas@wp.pl

**Keywords:** salivary biomarkers, stress, anxiety, depression

## Abstract

Stress, anxiety and depressive disorders are often characterized by the activation of the stress axis, which results in similar symptoms at some point in these disorders. These disorders are closely related to each other—they occur simultaneously or follow one another. The diagnosis of stress, anxiety and depression is not a perfect procedure currently—it is based on patient observation and an interview with the patient and their family. There are no laboratory tests that would dispel the doubts of the doctor making the diagnosis and allow the appropriate treatment to be implemented as soon as possible. Therefore, this study will review the components of saliva that could be helpful in the quick diagnosis of stress, anxiety and/or depression. Such potential salivary biomarkers could also be useful in monitoring the effectiveness of pharmacological treatment prescribed by a psychiatrist. The following are promising salivary biomarkers of stress, anxiety or depression: cortisol, immunoglobulin A (sIgA), lysozyme, melatonin, α-amylase (sAA), chromogranin A (CgA) and fibroblast growth factor 2 (FGF-2). To the best valuable potential salivary markers of stress, we can include cortisol, lysozyme, sAA and CgA. To differentiate depression from stress, salivary cortisol and melatonin can be helpful. Fluctuations in the concentrations of the above-mentioned substances in saliva indicate a particularly strong relationship with typical human psychological problems, such as stress, depression or anxiety.

## 1. Stress, Anxiety and Depression

For the past decade, more and more affective disorders have been diagnosed worldwide. Research reports suggest that depression is the second most common disease in the world after heart failure and may even take the lead in 2030 [1,2,3]. Depressive disorders are very often preceded by the stress and accompanied by anxiety—the coexistence of anxiety and depression is found in nearly 75% of children and adolescents [3,4]. However, if anxiety and depressive disorders do not occur at the same time, they usually follow one another. It is estimated that the probability of anxiety disorders after a depressive episode is 47–58%, and 56% of patients with anxiety disorders develop depression [5,6,7]. About 50% of depressive episodes are preceded by sudden, negative life events experienced by patients [8,9,10,11,12]. Other factors predisposing to a depressive episode include chronic somatic diseases, including diabetes, multiple sclerosis, neoplastic diseases, myocardial infarction, stroke, obesity, etc. [13]. In both of the above-mentioned cases, we deal with stressful situations. Experiencing stress definitely influences the emotional state of patients, as it initiates changes in the brain’s neurochemistry. Chronic stress causes a decrease in serotonin levels, changes in the density of serotonergic system receptors, an increase in glutamate levels and a decrease in the concentration of brain-derived neurotrophic factor (BDNF) [14]. Unfortunately, stress is a phenomenon that occurs in everyone’s life. Additionally, most people feel anxious or even depressed from time to time. If the symptoms listed above are mild and pass in a relatively short time, they do not pose any major health risks. What is more, experiencing temporary stress is a positive reaction for the body, thanks to which processes that enable a person to properly cope with the challenges faced are triggered [8,14]. The actual anxiety or depressive disorders manifest themselves in a long-lasting, self-limiting intensification of experiencing negative emotions. Anxiety and depression definitely make it difficult for the patient to function in society. Over time, a patient with anxiety or depression loses their self-esteem and, subsequently, loses the motivation to take any life actions. Patients suffering from anxiety and/or depression suffer from problems with sleeping, appetite, psychomotor activity (agitation or impairment), dysregulation of the autonomic nervous system and digestive tract, sometimes even from psychosis, which can lead to thoughts and even attempted suicide. Anxiety and depression are complex disorders that have many subtypes and etiologies [7,9,10,15,16]. Currently, in psychiatry, the main method used to diagnose stress, anxiety and/or depression are patient observation and medical interviews conducted with the patient and members of their immediate family (Figure 1). In the diagnosis of depression and/or anxiety, additionally, the so-called diagnostic tests (e.g., Beck and Montgomery-Asberg in depression, Hamilton in depression and anxiety and Perceived Stress Scale) are used [17,18,19]. The above-mentioned diagnostic tests are very valuable in the diagnosis of stress, anxiety and depression, but it would be better if they could complement the proper diagnosis or be used to assess the severity of the disorder or to observe the effectiveness of the therapy [14]. The multitude of causes of stress, anxiety and depression, as well as the symptoms overlapping with those of other mental disorders, often make it difficult for psychiatrists to make an appropriate diagnosis and apply effective treatment. There are no specific laboratory tests that would be helpful in the diagnosis of anxiety and depression.

Therefore, this study will review salivary biomarkers that may be helpful in the diagnosis of stress, anxiety and depression. We chose saliva because in our opinion saliva is an ideal biological fluid for diagnostic, because it does not coagulate, it is stable for 24 h at room temperature and for a week at 4 °C. The use of saliva in medical diagnostics is becoming more and more popular; it is cheap, non-invasive, painless and convenient for both young children and the elderly. Saliva can be collected at the patient’s home and transported to the laboratory without the involvement of any medical personnel [20,21]. Saliva contains many substances, the concentration of which reflects the health of the whole organism, which can be used in easy and quick detection of early pathological changes in humans [22,23,24,25]. Saliva components can be cheaply determined using specific and sensitive immunological and/or biochemical methods, such as radioimmunoassay (RIA), enzyme-linked immunosorbent assay (ELISA), spectrophotometry or chromatography [26,27].

Despite the advantages of saliva, which we have listed above, we are aware that saliva is not an ideal biological material. The individual composition of saliva can be influenced by many factors, such as age, sex, smoking, diet, medications or the pathophysiological condition of the oral cavity. There are also other biological materials that are also non-invasive and cheap to obtain, such as hair, but they also have some disadvantages, as the composition of the hair can be highly adulterated with hair dyes or other cosmetics used for daily care [28].

## 2. Stress Biomarkers

Under the influence of stressful stimuli, the physiological activation of specific areas of both the central and peripheral nervous system takes place. The nervous system’s response to stress includes stimulation of the hypothalamus and brainstem, activation of the hypothalamic–pituitary–adrenal (HPA) axis and the autonomic nervous system (ANS), which includes the efferent sympathetic–adrenal–medullary (SAM) system. These systems work together with the immune system and are closely involved in the pathogenesis of stress-related diseases [29]. All of the body’s compensatory responses listed above involve the production of organic chemicals that can subsequently be found in the blood serum and saliva. In the literature, you can find publications indicating the relationship between stress, depression and/or anxiety and the concentration of cortisol, immunoglobulin A, lysozyme, melatonin, alpha-amylase, chromogranin A and/or fibroblast growth factor 2 in saliva.

## 3. Cortisol

Cortisol is a hormone produced by the adrenal cortex, mainly in the second half of the night, which means that serum cortisol levels are highest between 7 a.m. and 8 a.m. During the day, the levels of cortisol drop significantly and, in the evening, only about 10% of the morning cortisol remains in the body. When a person is exposed to mental or physical stress, the adrenal glands produce increased amounts of cortisol. Cortisol activates metabolism, which provides the body with energy (e.g., by releasing glucose into the blood) and changes the conditions of mental reactions (by enhancing the action of other “stress hormones”—adrenaline and noradrenaline—thanks to which the body gets an additional boost of energy to cope with the situation stress) [30,31,32]. Therefore, cortisol is a substance most often indicated by researchers as a stress biomarker [33,34,35,36]. Miller et al. conducted a meta-analysis of the link between strong, short-term stress with the activity of the hypothalamic–pituitary–adrenal (HPA) axis and confirmed the importance of salivary cortisol as a biomarker of acute stress. They provided that when the stressor decreased, the morning level of cortisol in the saliva also fell down [37]. The stress is present nearly every day for almost everyone: at work, at school or at home. As a result of being under permanent stress, the human body is exposed to the effects of constantly high levels of cortisol. Therefore, there is a hypothesis that after long exposure to stress (both mental and physical), the HPA axis becomes less sensitive, which results in a decrease in the production of cortisol by the adrenal glands [38,39]. This theory is confirmed by the results of studies conducted with the participation of people who were accompanied by long-term anxiety disorders. It turned out that, paradoxically, they had a lower cortisol production than in people not under stress [40]. However, all obtained by the above-mentioned researchers’ results were mainly based on cross-sectional studies of one morning’s saliva sample. Yonekura et al. [41] analyzed the relationship between salivary cortisol levels in adolescents who survived the eastern Japanese earthquake. They found that it was difficult to distinguish people with symptoms of severe depression only on the basis of salivary cortisol point measurements. However, when they compared the changes in salivary cortisol levels over three consecutive days, they found that the group of adolescents with severe symptoms of depression had a significantly higher ratio of morning to evening levels of salivary cortisol compared to the group of adolescents with mild symptoms of depression. Yonekura et al., using the receiver operating characteristic (ROC) curve, showed that cortisol measurements repeated in saliva over three consecutive days can identify adolescents with depressive symptoms [41]. The use of salivary cortisol as a potential biomarker of stress and depression requires further research, because the concentration of cortisol in the saliva depends on the disturbance and the circadian rhythm of its secretion. Additionally, we should underline the fact that meta-analyses detected a large heterogeneity between the studies of salivary cortisol, and the study reported that these results seem to be due to lack of standardized kits for measuring salivary cortisol, clinical heterogeneity of the included participants and the bias of risk factors [42].

## 4. Immunoglobulin A (IgA)

Stress can weaken the immune system and inhibit the production of immunoglobulins [43]. IgA is a class of antibodies found in the mucous membranes, which is the body’s first defense factor in contact with an infectious pathogen or an allergen [43]. The level of IgA in the saliva also changes in response to psychological factors, such as positive (increase) and negative mood (decrease), or stressful stimuli (decrease) [43,44]. Animal studies have shown that IgA levels are reduced after both prolonged and acute stress [45]. It has been proven that chronic stress in humans is related to the activation of the HPA axis (measured by an increase in salivary cortisol concentration) and to decreased immune system activity (measured by a decrease in salivary IgA and lysozyme concentration) [43,46]. It was even concluded that there was a strong relationship between the perceived stress, depressive symptoms and anxiety with a low level of salivary immunoglobulin A [46].

## 5. Lysozyme

Lysozyme is a low-molecular-weight cationic protein that is constantly synthesized and released from monocytes and macrophages and is widely distributed in human tissues and secretions, including saliva. Lysozyme gives saliva antibacterial properties. It takes part in the host’s innate immunity defense against pathological microorganisms due to the fact that it has the ability to hydrolyze peptidoglycan β-1,4-glycosidic bonds of bacterial cell walls. Lysozyme also plays a key role in antiviral defense [47,48] and induces the lysis of neoplastic cells [47]. Perera et al. [49] found significantly lower concentrations of lysozyme in samples of saliva collected from students just before the exam compared to those obtained after the end of the exam session, proving that salivary lysozyme can be used as a potential stress marker. Yang et al. [50] showed that salivary lysozyme concentrations are lower in emergency nurses experiencing high occupational stress, compared to nurses who commonly care for patients in non-emergency wards. A negative correlation between the concentration of lysozyme and exposure to stress was also demonstrated by other researchers [51,52], which additionally confirm the theory that there was an increased susceptibility to any infections during exposure to stress. Currently, lysozyme is used as a biomarker in the diagnosis of chronic stress [52]. However, there is no information on the concentration of lysozyme in the saliva of people suffering from depression or anxiety.

## 6. Melatonin

Melatonin is a serotonin derivative, produced by pinealocytes, found in the pineal gland. The main task of melatonin is to synchronize the human internal biological clock with the day and night cycle. The concentration of melatonin in the body fluctuates throughout the day: it rises in the evening until it reaches its peak at 2–4 a.m. and then gradually decreases until the morning hours, when melatonin production is stopped. The interaction of melatonin with the main melatonin receptors modulates phases of sleeping, which ensures the quality of sleep [53]. Melatonin also has strong antioxidant properties. By enabling the proper regeneration of the human body during sleep, melatonin increases resistance to external factors (both physical and mental) [54]. Sleep disturbances can also be caused by a chronic stress [55]. Sleep disorders include difficulty falling asleep, inability to maintain continuous sleep, resulting in poor quality of sleeping or waking up prematurely. Long-term sleep problems can have a negative effect on health by lowering immunity, increasing the risk of high blood pressure, heart disease or insulin resistance. Circadian rhythm disorders (including insomnia), which often occur during stress, depression or anxiety, additionally increase the level of mental stress and deepen the depressive state [55,56]. Voultsios et al. [57] proved that the level of salivary melatonin is strongly correlated with the level of melatonin in the plasma. Therefore, based on the concentration of melatonin in the saliva, circadian rhythm of its secretion can be assessed. Ito et al. [58] reported that the level of salivary melatonin (both nighttime and daytime) was strongly correlated with depression and, therefore, suggested that salivary melatonin could be successfully used as a salivary biomarker in the diagnosis of depression. Moreover, the same authors proved that melatonin levels in saliva at night are moderately correlated with the anxiety disorders [58]. However, that correlation was not as strong as in the case with depression. Paul et al. [59], in turn, found a significant reduction in the levels of salivary melatonin at night in soldiers suffering from post-traumatic stress disorder (PTSD). The results published by researchers seem to be very promising. Therefore, melatonin in the saliva may become a clinically useful biomarker in the future [60].

## 7. Salivary Alpha-Amylase (sAA)

Salivary alpha-amylase (sAA) is the main digestive enzyme of the oral cavity, which, apart from the hydrolysis of starch and glycogen, also has an immunological function, protecting the oral cavity from microorganisms [61]. Salivary alpha-amylase has also been recognized as a marker that is sensitive to stimuli activating the sympathetic system (e.g., adrenaline) [62]. In health, alpha-amylase levels are the lowest in early morning and the highest in late afternoon [63]. In response to stress, the concentration of alpha-amylase in saliva increases abruptly, which makes it an important salivary biomarker of stress [64], particularly acute stress, e.g., during blood sampling [65]. Acute stress activates the axis of sympathetic nervous system, adrenal medulla, which is reflected not only in the concentration of salivary alpha-amylase, but also in the concentration of salivary chromogranin A [46]. The usefulness of salivary alpha-amylase as a salivary biomarker of acute stress and anxiety was confirmed by Jafari et al. [66] pointing to its reliability and objectivity in the measurement of anxiety associated with dental treatment. In turn, Vineetha et al. [67] showed that alpha-amylase activity in the saliva increases in patients experiencing chronic stress. Van Veen et al. [68], however, suggested that sAA could be a potential salivary stress marker in people sensitive to a negative social assessment.

## 8. Chromogranin A (CgA)

CgA is an acidic glycoprotein released from the adrenal medulla and sympathetic nerve endings that can be measured in the saliva [69,70]. A significant increase in the concentration of CgA was found in the saliva of pigs subjected to isolation stress or stress related to immobilization, thus suggesting salivary chromagranin A as a sensitive and specific biomarker of acute stress in pigs [71]. Salivary CgA is also indicated by many researchers as a promising, sensitive salivary biomarker of psychosomatic stress in humans [72,73,74,75,76,77,78]. It has been proven that a rapid and specific increase in CgA in human saliva occurs in response to various stress factors, e.g., public speaking [72], sports [79], noise [75,80], passing an academic exam [81], medical intervention [76,77,82] or driving a car [75]. However, there are no reports on the level of salivary chromogranin A in people diagnosed with depression or anxiety. A significant advantage of determining salivary CgA, compared to the above-mentioned biomarkers of stress in saliva (cortisol, alpha-amylase, IgA), is that CgA levels are not affected by the time of the day. Although concentrations of salivary CgA peak shortly after waking up, they decline very quickly (within only 1 h), remaining constant throughout the day [72].

## 9. Fibroblast Growth Factor 2 (FGF-2)

Fibroblast Growth Factor 2 (FGF-2) is a mitogen for various cell types found in the salivary glands and saliva involved in a number of physiological activities related to neuroregeneration and stress regulation [83,84]. FGF-2 has been recognized as an endogenous regulator of fear expression. The results of studies carried out on rats indicate that FGF-2 is a promising biomarker of susceptibility to stress and anxiety disorders [85,86]. In studies carried out in a large group of healthy people who were subjected to various stress stimuli, salivary FGF-2 levels correlated negatively with the expression of fear/stress, regardless of the type of stimuli acting on them [87]. FGF-2 can, therefore, be used as a biomarker of stress. Salivary glands, which are reservoirs for growth factors, including FGF-2, can also secrete FGF-2 in response to stress. Acute stress also transiently increases hippocampal FGF-2 in rats [88]. People with reduced FGF-2 reactivity in response to stress may have a reduced physiological ability to cope with stress, which places them at an increased risk of developing anxiety [84].

## 10. Others Biochemical Parameters

Some authors also point to the role of other salivary biochemical substances as potential biomarkers of stress, anxiety and/or depression, e.g., steroid hormones (testosterone, dehydroepiandiosterone sulfate—DHEA-S, aldosterone) [89,90]. However, in both the human and animal models, there are many factors, such as environment, age and, above all, sex, that contribute to significant differences in the concentrations of individual steroid hormones in anxiety and depressive disorders. [91]. Therefore, we have chosen not to consider these substances as potential salivary biomarkers of stress, anxiety and/or depression in this review.

## 11. Summary

With the presented medical knowledge, our suggestions are preliminary—most of the salivary biomarkers presented in this paper show only potential applicability as markers of stress, anxiety and depression. To the best valuable potential salivary markers of stress, we can include cortisol, lysozyme, sAA and CgA (Table 1). It would be advisable to conduct additional analyses in order to try to develop the so-called assay profiles, separate for specific mental disorders, which would define concentration/activity ranges of individual markers in such a way that they could be used to distinguish between acute and chronic stress, anxiety and depression. It also seems necessary to determine the optimal time of day for sampling, as some of the substances described above show a circadian rhythm.

Moreover, due to the fact that there are many reports that indicate the potential role of other biochemical substances as markers of stress, anxiety and/or depression in serum, plasma, urine or even hair, it would also be advisable to perform laboratory experiments and determine whether there is a correlation between individual parameters present in other biological materials and their content in saliva, e.g., malondialdehyde [92] or nesfatin-1 [93].

## Figures and Tables

**Figure 1 jcm-10-00517-f001:**
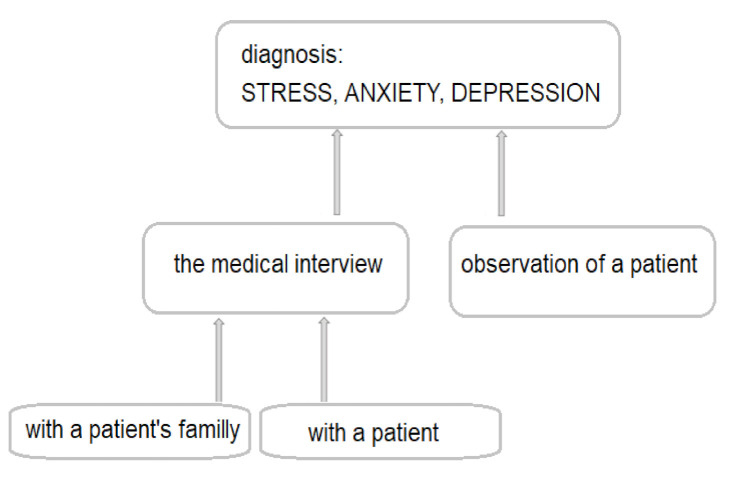
The procedure currently used to diagnose stress, anxiety and depression.

**Table 1 jcm-10-00517-t001:** The salivary biomarkers of stress, anxiety and depression and their daily variability. ↑—increased concentration/activity, ↓—decrease concentration/activity, ?—inconclusive or no results, *—scientifically proved/the most probable markers.

Salivary Biomarker	Acute Stress	Chronic Stress	Anxiety	Depression	Daily Variability
Cortisol	↑ * [Heim et al. 2000] [38]	↓ [Miller et al. 2007] [37]	↓ [Hek et al. 2013] [40]	morning levels > evening levels [Yonekura et al. 2014] [41]	+ [Hellhammer et al. 2009] [33] [Yonekura et al. 2014] [41]
Immunoglobulin A (sIgA)	↓ [Svobodova et al. 2014] [45]	↓ [Svobodova et al. 2014] [45] [Cohen et al. 2001] [43] [Nomura et al. 2011] [44]	↓ [Engeland et al. 2016] [46]	↓ [Engeland et al. 2016] [46]	- [Engeland et al. 2016] [46]
Lysozyme	↓ [Perera et al. 1997] [49]	↓* [Yang et al. 2002] [50]	?	?	- [Perera et al. 1997] [49]
Melatonin	?	↓ [Paul et al. 2019] [59]	↓ [Ito et al. 2013] [58]	↓ * [Ito et al. 2013] [58]	+ [Voltious et al. 1997] [57] [Ito et al. 2013] [58]
Alpha-amylase (sAA)	↑ * [Takai et al. 2004] [62] [Strahler et al. 2010] [64] [Koh et al. 2014] [65]	↑ [Vineetha et al. 2014] [67]	↑ [VanVeen et al. 2008] [68] [Jafari et al. 2018] [66]	?	+ [O’Donnell et al. 2009] [63]
Chromogranin A (CgA)	↑ * [Nakane et al. 1998] [72] [Nakane et al. 2002] [73] [Kanamura et al. 2006] [74] [Obara et al. 2005] [76] [Lee et al. 2006] [79] [Miyakawa et al.2006] [80] [Takatsuji et al. 2008] [81]	?	?	?	- [Kanamura et al. 2006] [74] [Takatsuji et al. 2008] [81]
Fibroblast Growth Factor 2 (FGF-2)	↓ [Bland et al. 2006] [87] [Duits et al. 2015] [86] [Graham et al. 2017] [85]	↓ [Bland et al. 2006] [87] [Duits et al. 2015] [86] [Graham et al. 2017] [85]	↓ [Bland et al. 2006] [87] [Duits et al. 2015] [86] [Graham et al. 2017] [85]	?	- [Graham et al. 2017] [85]

## Data Availability

Data sharing not applicable.

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
