# Peer review of "Salivary Biomarkers of Stress, Anxiety and Depression"

_jcm, 2021, doi:10.3390/jcm10030517_

Round 1

Reviewer 1 Report

This is a work that provides valuable information for the empirical diagnosis of pathologies on which the diagnosis is often based on subjective functions of the patient.

  1. It would be very useful to include a table of references in the article in order to facilitate the reader's understanding of the text. The table can include several columns where you can indicate author, fluid studied, type of study, etc.
  2.  There are no previous studies of most of the markers included in the study and their relationship with anxiety and depression pathologies. For this reason, it would be of interest to establish a correlation between the markers in which there are no data in saliva, although there are in other fluids such as serum and plasma. For example, in reference to cortisol levels and its relationship with anxiety and depression pathologies, we have some studies that relate them in plasma and serum: Islam M. R., Islam M. R., Ahmed I., Moktadir A. A., Nahar Z., Islam M. S., et al. . (2018). Elevated serum levels of malondialdehyde and cortisol are associated with major depressive disorder: a case-control study. SAGE Open Med. 6:2050312118773953. 10.1177/2050312118773953. SERUM Xu Y. Y., Ge J. F., Liang J., Cao Y., Shan F., Liu Y., et al. . (2018). Nesfatin-1 and cortisol: potential novel diagnostic biomarkers in moderate and severe depressive disorder. Psycho. Res. Behav. Manag. 11, 495–502. 10.2147/PRBM.S183126. PLASMA The same can be seen with the marker "lysozyme", in which I do not detect conclusive information for anxiety and depression.

Author Response

We thank the Reviewer #1 for their careful reading of the manuscript and their constructive remarks. We have taken the comments on board to improve and clarify the manuscript. Please find a detailed point-by-point response to all comments in the attached file (reviewers' comments in black, our replies in blue).

Reviewer 2 Report

This study indicates that some saliva biomarkers that could be helpful in the quick diagnosis of stress, anxiety and/or depression.

I think the theme of this paper is suitable for publication in Journal of JCM, and agree the using of useful biomarker would be helpful in the diagnosis of anxiety and depression. However, a number of concerns for this manuscript could be mentioned.

  1. This review does not seem to present new scientific advances or interpretations in that field. Throughout, the current references and presented biomarkers should be updated. For example, maybe you should add testosterone, DHEA-S, or aldosterone.

  1. Meta-analyses detected a large heterogeneity between the studies of salivary cortisol, and the study reported that these results seems to be due to lack of standardized kits for measuring salivary cortisol, clinical heterogeneity of the included participants, and the bias of risk factors (Knorr et al., 2010). Thus, if you also indicate about the factors which effect to using the biomarkers, it may useful for the future investigation by the readers using that biomarkers (e.g., effect of sampling point, age, and gender).

  1. Are there evidences in the Table 1 or in the current review that to differentiate depression from stress salivary cortisol and melatonin seem to be helpful? In addition, the summary is not well-founded, so it is advisable to reconsider the text to support your results. Alternatively, I suggest modifying the summary table to an accurate table based on overall reviews. Although current table is simple and easy to look at, I feel this presentation is lacking or inaccurate.

  1. In the Introduction, as a limitation of saliva samples, you could describe comparison with other samples (e.g., urine and hair), intra-individual stability.

I hope these comments will be helpful.

Author Response

We thank the Reviewer #2 for their careful reading of the manuscript and their constructive remarks. We have taken the comments on board to improve and clarify the manuscript. Please find a detailed point-by-point response to all comments in the attached file (reviewers' comments in black, our replies in blue).

Round 2

Reviewer 2 Report

I got positive responses from authors to my previous concerns. However, some major and minor concerns still remain.  

Major concerns:

  1. I am still not convinced that the summary discussion is based on the Table 1. You describe that "the current review that to differentiate depression from stress salivary cortisol and melatonin seem to be helpful" (Page 3, L92). Why can you provide such interpretation?  Please explain. I would suggest to be more careful with interpreting this summary or to add persuasive and easy to understand discussion.

  1. You describe that urine is not as biologically stable as saliva (Page 3, L92). To me it seems this explanation is not suitable here. For example, although urine and hair showed moderate test-retest correlations and good multiple-test consistencies, three single-point salivary biomarkers showed weak to moderate test-retest correlations and poor to fair multiple-test consistencies within two weeks (Quan Zhang et al., 2017). Please be more careful and consider rewriting this into a more logical manner. I would advise the authors to rewrite the review on a neutral position.

Minor points:

  1. Summary ⇒ 11. Summary (Page 6, L265)

Author Response

We would like to thank Reviewer #2 for the constructive and competent criticism. Modified sections in the revised version of manuscript are highlighted in green. In the attached file we provide a detailed point-by-point reply.
